# Plasma-Etched Vertically Aligned CNTs with Enhanced Antibacterial Power

**DOI:** 10.3390/nano13061081

**Published:** 2023-03-16

**Authors:** Emily Schifano, Gianluca Cavoto, Francesco Pandolfi, Giorgio Pettinari, Alice Apponi, Alessandro Ruocco, Daniela Uccelletti, Ilaria Rago

**Affiliations:** 1Dipartimento di Biologia e Biotecnologia “C. Darwin”, Sapienza University of Rome, Piazzale Aldo Moro 5, 00185 Rome, Italy; 2SNN Lab, Sapienza Nanotechnology & Nano-Science Laboratory, Sapienza University of Rome, 00100 Rome, Italy; 3Dipartimento di Fisica, Sapienza University of Rome, Piazzale Aldo Moro 2, 00185 Rome, Italy; 4INFN Sezione di Roma, Piazzale Aldo Moro 2, 00185 Rome, Italy; 5Istituto di Fotonica e Nanotecnologie, CNR-IFN, Via del Fosso del Cavaliere 100, 00133 Rome, Italy; 6Dipartimento di Scienze, Università Degli Studi Roma Tre and INFN Sezione di Roma Tre, Via della Vasca Navale 84, 00146 Rome, Italy

**Keywords:** carbon nanotubes, chemical vapor deposition, antimicrobial, plasma etching, nanomorphology

## Abstract

The emergence of multidrug-resistant bacteria represents a growing threat to public health, and it calls for the development of alternative antibacterial approaches not based on antibiotics. Here, we propose vertically aligned carbon nanotubes (VA-CNTs), with a properly designed nanomorphology, as effective platforms to kill bacteria. We show, via a combination of microscopic and spectroscopic techniques, the ability to tailor the topography of VA-CNTs, in a controlled and time-efficient manner, by means of plasma etching processes. Three different varieties of VA-CNTs were investigated, in terms of antibacterial and antibiofilm activity, against *Pseudomonas aeruginosa* and *Staphylococcus aureus*: one as-grown variety and two varieties receiving different etching treatments. The highest reduction in cell viability (100% and 97% for *P. aeruginosa* and *S. aureus*, respectively) was observed for the VA-CNTs modified using Ar and O_2_ as an etching gas, thus identifying the best configuration for a VA-CNT-based surface to inactivate both planktonic and biofilm infections. Additionally, we demonstrate that the powerful antibacterial activity of VA-CNTs is determined by a synergistic effect of both mechanical injuries and ROS production. The possibility of achieving a bacterial inactivation close to 100%, by modulating the physico-chemical features of VA-CNTs, opens up new opportunities for the design of self-cleaning surfaces, preventing the formation of microbial colonies.

## 1. Introduction

Antibiotics, because of their affordable costs and quick action, are the main treatment of bacterial infections. In response, bacteria have developed the ability to evolve rapidly through mutations, becoming resistant to these treatments. In addition, bacterial species can transfer drug-resistant genes to each other through horizontal gene transfer, which is the acquisition of foreign genes by an organism [1]. This results in the emergence of multidrug-resistant (MDR) bacteria [2], an increasing trend due to the amplified use of antibiotics, especially during the coronavirus pandemic [3]. This adaptive response is linked to the microbes’ ability to adapt and survive in the most diverse conditions, often in the form of biofilms.

Biofilms are colonies of microorganisms enclosed in extracellular polymeric substances (EPSs) attached to a surface [4]. Normally, the reversible attachment of planktonic (free-floating) bacteria on a substrate becomes irreversible in the presence of favorable conditions. At first, cells proliferate and aggregate to form microcolonies, and then they create robust biofilms that are more resistant to antibiotic and antiseptic treatments [5]. The surrounding EPS physically protects these microorganisms in hostile environments, upregulating and exchanging genes responsible for producing antimicrobial resistance. To date, strategies that can effectively prevent bacterial adhesion to avoid the development of biofilms are highly demanded.

A promising approach to prevent the microbial colonization of surfaces turned out to be the use of micro- and nano-topographies, without involving the release of any biocidal agent to which pathogenic bacteria can develop resistance after a prolonged exposure [6,7]. The proposed killing mechanism for these antibacterial micro- and nano-tools has been depicted as a physical mechano-bactericidal action between the bacterial cell walls and the micro-/nano-structured surfaces [8]. Most of the models developed to figure out the dynamic networking between microorganisms and nanopatterned substrates have highlighted a bactericidal process that, depending on the shape of these nano-features, takes place via the stretching or cutting/piercing of the bacterial cell membrane [6,9,10,11,12,13]. In particular, nanopillar arrays have revealed their capability to induce bacterial membrane stretching [14,15], while sharp nano-edges have been reported to be responsible for cell wall damage via punctuations [16,17,18,19]. For both these nano-engineered substrates, it was observed that the mechano-bactericidal efficiency is strongly affected by their geometric nano-features (i.e., aspect ratio, spacing, and curvature radius) [20,21,22,23,24], thus suggesting that the fine-tuning of these characteristics at the nanoscale represents a valuable approach to modulate their antibacterial activity.

Among the most effective antimicrobial nanomaterials, carbon nanotubes (CNTs) have been extensively investigated, demonstrating their strong bactericidal properties toward both planktonic (free-floating bacteria) and biofilm (bacterial community) infections [25,26,27]. Most of the studies on the antibacterial properties of CNTs relate to their randomly oriented configuration and, in particular, to CNT-based suspensions. Despite their ascertained bacteria-killing effects [28,29,30,31], the use of CNTs dispersed in media still exhibits limitations, including the tendency to aggregate if not properly functionalized, which can reduce their antibacterial performances [32]. More critically, in the case of floating CNTs the results related to their cytotoxicity are conflicting, because they spread from negligible cellular reactions to severe toxic effects influenced by CNT length, diameter, concentration, and chemical functionalization [33,34,35]. However, when CNTs were strongly anchored on substrates, cytotoxicity consequences were not highlighted [36,37,38]. Notably, in order to tailor the antibacterial capacity of these carbon nanostructures, it is crucial to shape the spatial arrangement up to the sub-micrometric domain. In this regard, while CNTs dispersed in liquid media are difficult to control, vertically aligned carbon nanotubes (VA-CNTs), also known as CNT forests, can offer a tunable antibacterial platform thanks to the possibility to direct the tubes’ alignment, diameter, length, and density via both in and ex situ processes [39,40]. The catalytic chemical vapor deposition (CVD) technique is the most widely exploited process for the synthesis of VA-CNTs [41]. Such an approach allows, in the presence of a carbon precursor (i.e., hydrocarbon source gas), the growth of highly oriented CNTs on solid substrates covered by a nanostructured catalytic element (i.e., Fe, Co, and Ni nanoparticles). In CVD synthesis, both catalyst characteristics (i.e., type, size, and density) and process parameters (i.e., growth temperature, reaction pressure and time, and gas composition and flow rate) dictate tube morphology and structural properties [42,43]. At the onset of CVD growth, straight and curved tubes are formed at the same macroscopic rate due to the interaction between fast- and slow-grown CNTs, thus leading to the self-arrangement of tubes in an entangled and randomly oriented configuration, known as the “crust layer” [44]. After the formation of such an intricate network of CNTs, the tubes continue to grow perpendicularly to the underlying substrate due to van der Waals interactions between them [45,46], displacing the crust on top of the forest [40].

The possibility of finely shaping this CNT-based crust, without damaging the microstructural integrity of the underlying VA-CNT forest, is crucial from the perspective of engineering the antibacterial activity of CNTs. Various strategies have been proposed to achieve this goal, including laser ablation [47], liquid-phase oxidation cutting, and electron beam and focused ion beam (FIB) processes [48]. However, these techniques can alter CNTs’ features and alignment [49]. Alternatively, as in the case of FIB, a high level of precision in modifying the tube morphology can be achieved, but at the same time, undesired by-products and carbon redeposition are formed, and, in addition, long manufacturing times are required [50]. Conversely, plasma treatments have largely demonstrated to be capable of tailoring the morphology of CNTs in a controlled and time-efficient manner [40,51,52].

Herein, we firstly demonstrate that uniform, highly aligned, and densely packed CNTs can be successfully synthesized on SiO_2_/Si substrates via a rapid (~15 min), cost-effective, and versatile synthesis method developed in a customized CVD system. The morphological (i.e., vertical alignment, density, and uniformity) and topographical (i.e., roughness, diameters, and height profiles) features of the as-grown highly oriented carbon nanostructures are assessed via field-emission scanning electron and atomic force microscopies (FE-SEM and AFM), while their surface composition is studied via X-ray photoelectron spectroscopy (XPS). Afterwards, the ability to tailor the surface morphology of VA-CNTs by means of plasma etching processes is exploited to systematically investigate the antimicrobial properties of different CNT-based substrates, with properly designed surfaces, against both Gram-positive (*Staphylococcus aureus*) and Gram-negative (*Pseudomonas aeruginosa*) bacteria.

## 2. Materials and Methods

### 2.1. CVD Synthesis of VA-CNTs

Vertically aligned carbon nanotubes were grown at the INFN CVD facility in Sapienza University of Rome by employing a customized thermal CVD reactor with a base pressure in the low 10^−7^ mbar range. This CVD facility has been developed for various applications, including the development of a novel dark matter detector [53,54,55]. Silicon p-type/Boron-doped < 100 > SiO_2_/Si wafers, here adopted as growth substrates, were firstly cleaved into 40 mm × 20 mm chips and, subsequently, cleaned via the Radio Corporation of America (RCA) method. Electron beam (E-beam) evaporation was exploited for the deposition of a thin (3 nm in thickness) catalyst layer of iron over the Si-based substrates, which were then mounted on a heating element inside a high-vacuum reaction chamber and annealed at 720 °C in a H_2_ atmosphere for 4 min. The purpose of such thermal treatment is twofold: firstly, to reduce iron oxides possibly developed on the surface of the Fe/SiO_2_/Si substrate because of the exposure to the atmospheric air occurring when the sample is transferred from the E-beam evaporator to the CVD system and, secondly, to trigger the de-wetting of the catalyst layer and the nucleation of iron-based nanoparticles working as a template for subsequent CNT growth. After the annealing treatment, the reaction temperature was increased to 740 °C, and acetylene (carbon precursor) was fluxed inside the CVD chamber at a 300 sccm flow rate, without any carrier gas, up to a partial pressure of about 50 mbar. The growth time, intended as the time of interaction between the iron nanoparticles and the carbon source, was limited to 10 min. The optimization of the process parameters was based on previous studies related to the synthesis of VA-CNTs on different substrates [36,37,38,43]. Subsequently, the sample temperature was decreased to room temperature under the base pressure of the CVD system.

### 2.2. Plasma Etching Treatment

The as-grown VA-CNTs were subjected to O_2_ and Ar/O_2_ plasma treatments in a Plasmalab 80 plus reactive ion etching system from Plasmalab Technology (Everett, WA, USA). The etching studies were stimulated by a previous study [40], which first showed the possibility to remove the crust layer from CNT forests by means of Ar/O_2_ plasma treatments. Here, we investigated different plasma etching conditions to tailor the crust layer by maintaining the typical high verticality of our VA-CNTs. CNT forests were subjected to plasma etching with different RF powers (from 33 to 100 W), different Ar (from 0 to 52 sccm) and O_2_ (from 2 to 195 sccm) flow rates, different residual pressures (from 1 to 200 mTorr), and different etching times (from 1 to 5 min) (Appendix A). The effect of the plasma treatment on the VA-CNTs’ morphology was evaluated by means of SEM, AFM, and XPS characterizations, and two different optimal etching conditions were finally selected: one with a combination of Ar and O_2_ gas etching (10 sccm of Ar, 2 sccm of O_2_, 35 W RF power, 200 mTorr residual pressure, 5 min of total etching time) and one with only O_2_ gas etching (66 sccm of O_2_, 100 W RF power, 40 mTorr residual pressure, 1 min of total etching time). The effects of these two conditions on the CNT morphology, roughness, surface composition, and antibacterial properties are reported below.

### 2.3. Characterization of VA-CNTs

FE-SEM imaging was conducted on the CNT forests, before and after the plasma etching processes, by collecting secondary electrons on a Zeiss Auriga SEM (Jena, Germany) operating at an accelerating voltage of 5 kV. Topographic measurements were performed with a Park Systems NX10 AFM (Suwon, Republic of Korea) in tapping mode, in air, by using a Nanosensors PPP-NCHR AFM tip (Neuchatel, Switzerland) and by collecting micrographs of 512 × 512 pixels at a cantilever speed of about 0.1 Hz. The surface composition of the VA-CNTs was investigated via an XPS analysis. An Omicron XM1000 monochromatized Al K_α_ X-ray source (Uppsala, Sweden) and a hemispherical electron analyzer (66 mm radius) equipped with a position-sensitive detector for parallel acquisition were employed. The energy of the X-rays was 1486.7 eV, and the total energy resolution was 520 meV. The C 1s spectra were fitted with Voigt profiles in order to take into account the experimental resolution and the intrinsic line width. The binding energy scale was calibrated with a clean sample of highly oriented pyrolytic graphite (HOPG) by setting the binding energy of the C1s core level to 284.5 eV [56].

### 2.4. Bacteria Growth Conditions and Cell Viability Test

The bacterial *Staphylococcus aureus* ATCC 25923 and *Pseudomonas aeruginosa* ATCC 15692 strains used in this study were grown in Luria–Bertani (LB) broth at 37 °C overnight under shaking. For cell viability tests, treated or untreated specimens, after 30 min of UV sterilization, were drop-casted with 15 µL of a bacterial suspension (1 × 10^8^ cell/mL). Cells were recovered after 30 min, 4 h, and 24 h of incubation at 25 °C by submerging the substrates in a sterile tube with 2 mL of sterile H_2_O_dd_ and shaking for 2 min. The extent of bacterial survival was assessed using the colony count method (Colony Forming Unit, CFU) by spreading the diluted samples (1 × 10^−2^ and 1 × 10^−3^ cell/mL) onto LB agar plates incubated at 37 °C overnight. The SiO_2_/Si wafer samples w/o CNTs were used as a reference (untreated). The experiments were repeated three times, with three replicates for each treatment.

### 2.5. Biofilm Formation

For a biofilm production assay, an amount of 1 × 10^7^ cells/mL of *S. aureus* or *P. aeruginosa* was inoculated in 3 mL of LB on a 35 mm Petri plate, in which different types of CNT samples (1 cm × 1 cm) were separately submerged. The plates were then incubated for 24 h at 37 °C. As a control, the bacterial strains were inoculated on plates where SiO_2_/Si substrates (untreated samples) were soaked. After treatment, the cultures were gently removed, and the specimens were washed twice with sterile water and fixed for an SEM analysis.

### 2.6. Bacteria SEM Analysis

The investigated bacteria were fixed on the Si substrates, covered or not with CNTs, with 2% glutaraldehyde in sterile ultrapure H_2_O for 1 h at room temperature in the dark. After 2 washes in sterile ultrapure H_2_O (15 µL each), the samples were then dehydrated with sequential treatment with 30, 50, 70, 80, 90, and 96% ethanol for 5 min each. An SEM microscopy of cells, interfaced or not with CNTs, was performed with a Zeiss Sigma 300 field-emission gun (Carl Zeiss S.p.a., Milano, Italy) operating at an accelerating voltage of 0.3 kV.

### 2.7. Reactive Oxygen Species Evaluation

For reactive oxygen species (ROS), treated or untreated specimens, after 30 min of UV sterilization, were drop-casted with 90 µL of a bacterial suspension (1 × 10^9^ cell/mL). Cells were recovered after 2 h of incubation at 25 °C by submerging the substrates in a sterile tube with 1 mL of sterile H_2_O_dd_ and shaking for 2 min. 

Then, 100 µL aliquots from each sample were transferred into wells of a 96-well microtiter plate containing 100 µL of a 50 μM 2′,7′-dichlorofluorescein diacetate H_2_DCFDA fluorescent probe (Sigma-Aldrich, Milan, Italy). Fluorescent signals were read immediately and after 60 min by using a microplate reader at excitation/emission wavelengths of 485 and 520 nm. The initial readings were subtracted from the final readings, and the mean fluorescence was calculated from the triplicate. The results are expressed as the ROS levels relative to those of the untreated specimens (UT) and are the mean of three independent experiments.

## 3. Results

An SEM analysis was conducted on the VA-CNTs, before and after the plasma etching treatments, to assess the impact of these post-growth processes on the tubes’ morphology, alignment, and microstructural integrity. The optimization of the plasma etching parameters led to the identification of three main VA-CNT-based configurations, namely, light, modest, and aggressive etching (Appendix A), as suggested in [40], and here, they are indicated as as-grown, O_2_-etched, and Ar/O_2_-etched CNTs, respectively. A side-view SEM micrograph (Figure 1A) of the as-grown CNT forests revealed the presence of highly aligned and packed nanotubes, homogeneously covering the Si substrate, while, from the top view (Figure 1B, a higher magnification in the top-right inset), the CNTs appear with no preferential orientation. Such a uniform layer of randomly oriented CNTs, the “crust layer” formed during the early stages of the growth process [57], switches to a more spiky CNT arrangement after the O_2_ plasma etching treatment (Figure 1D).

The sharper nanotube array, also shown at a higher SEM magnification (Figure 1D, in the top right indicated with an asterisk), does not involve a perturbation of the vertical alignment of the underlying CNTs (Figure 1C). Additionally, the CNTs were exposed to the Ar/O_2_ plasma etching in order to increase their active surface by exploiting the combination of ion bombardment and the chemical reaction provided by the Ar and O_2_ plasma etching, respectively [58,59]. In the side-view SEM image reported in Figure 1E, it is possible to appreciate that the configuration of the CNT forests totally changes with respect to that of the as-grown and O_2_-plasma-treated CNTs. Indeed, the Ar/O_2_ plasma etching causes a microstructural modification of the CNTs, leading them to arrange into micro-pillar arrays uniformly distributed over the Si substrate. The top-view SEM image (Figure 1F) shows a less rough topography when compared to both the as-grown (Figure 1B) and O_2_-plasma-etched (Figure 1D) forests.

The effect of the plasma etching on the CNTs’ surface roughness, modified or not via plasma etching processes, was investigated via AFM to obtain a quantitative indication of the morphological variations that emerged from the SEM analysis. In particular, the RMS of the surface profile, as measured using AFM, was taken as an estimator of the surface roughness. Figure 2A shows the typical 3D topography of the as-grown CNTs exhibiting a surface roughness of 141.2 nm. Such an entangled fractal-like structure of nanotubes, deriving from their randomly oriented terminal part, was further pointed out via the SEM examination (Figure 2D). After the O_2_ plasma etching treatment, a slight increase in terms of roughness (187.4 nm) was observed (Figure 2B), possibly ascribed to the formation of CNT aggregates assuming a configuration sharper than that highlighted in the top-view SEM micrograph reported in Figure 2E. However, after carrying out the Ar/O_2_ plasma etching, a significant reduction in the top surface roughness (41.2 nm) was measured via AFM (Figure 2C), suggesting the potential removal of the crust layer, also identified in the SEM analysis (Figure 2F).

To further investigate the surface chemical reaction occurring between the CNT forests and O_2_ or Ar/O_2_ etching gas, XPS characterization was performed. The XPS survey spectra, regardless of the plasma etching treatment, revealed the presence of two main elements: carbon (C1s) and oxygen (O1s) (Figure 3A). The C1s core level of the as-grown VA-CNTs (Figure 3B) pointed out the most intense peak (77%) at 284.3 eV assigned to sp^2^-hybridized carbon atoms, along with a smaller component (6%) at 284.8 eV associated with the sp^3^-like configuration and intrinsically related to CNT defects [60,61]. Additionally, together with the typical broad peak at 290.8 eV due to the 𝜋-plasmon, residual oxygen contamination (5%) was measured. After the O_2_ and Ar/O_2_ plasma etching (Figure 3C,D), only minor changes in the C1s core level spectra were detected: in both cases, the most intense peak was associated with sp^2^-hybridized carbon (80% and 91%, respectively). The sp^3^-like configuration was higher for the sample plasma-etched with O_2_ (9%) than for the sample plasma-etched with Ar/O_2_ (3%). Finally, the residual oxygen contamination was 5% and 1%, respectively, for the two plasma treatments.

As a representative of Gram-positive bacteria, *S. aureus* survival was evaluated by performing a CFU counting analysis after incubation with different CNT specimens. After only 30 min of treatment, a significant reduction of 97% in cell viability was observed in the presence of the Ar/O_2_-etched CNTs with respect to that of the untreated cells. In the case of the as-grown and O_2_-treated CNTs, notable viability reductions of about 90% and 70%, respectively, were highlighted after 4 h of treatment (Figure 4A). The interaction between the bacteria and the nanostructures was investigated through an FE-SEM analysis: the bacterial cell wall showed mechanical damages caused by direct contact with the nanotubes, which formed a network around the bacterium body, triggering cellular collapse (Figure 4B,C). This network became denser in the case of the Ar/O_2_-treated CNTs, suggesting that the higher mortality observed in the cell viability tests could be due to the strong interactions with these nanostructures (Figure 4D). When testing the Gram-negative *P. aeruginosa*, similar behavior was observed. In particular, an almost total reduction (100%) of viability was observed after only 30 min of treatment with the Ar/O_2_-etched CNTs (Figure 5A). When the cells were treated with the as-grown or O_2_-treated CNTs, a reduction of 80% was detected after 4 h of treatment, and it reached 99.99% after 24 h. Moreover, in this case, through the FE-SEM analysis, it seems clear that cellular activity is extinguished by the nanotubes, which skewer or smother the bacterial cells (Figure 5B–D).

In the environment, microorganisms exist in the form of biofilms, made up of a group of cells that produce an extracellular matrix, allowing the bacterial population to resist adverse abiotic conditions [62]. For this reason, different CNT specimens were tested on cells’ ability to produce biofilms (Figure 6). The results confirm the mechanical damage triggered by CNTs: the untreated cells were intact, showing their characteristic shape (round and rod-shaped morphology for *S. aureus* and *P. aeruginosa*, respectively). Conversely, in the treated cells, the bacterial surface showed mechanical injuries caused by direct contact with the CNTs, which adhered to the cell wall, causing its breakdown. After the incubation of the cells with the Ar/O_2_-etched CNTs, interestingly, it was also possible to notice a strong reduction in the cell number in the observation field as compared to that in the untreated samples.

To investigate the mechanisms of action of the different types of CNTs, bacterial oxidative stress was evaluated through a ROS accumulation analysis. Interestingly, the treatment of *S. aureus* with the VA-CNTs induced ROS production levels reaching 30% and 60% after incubation with the as-grown and Ar/O_2_-etched CNTs, respectively (Figure 7A). However, the O_2_-etched CNTs did not induce variations in ROS levels. In the case of *P. aeruginosa*, all VA-CNTs could generate high levels of ROS, with a fraction of positive cells ranging from 20 to 50% (Figure 7B). In general, the Ar/O_2_-etched CNTs resulted in the highest oxidative potential.

## 4. Discussion

In this work, the antibacterial activity of CVD-grown VA-CNTs, modified or not with different plasma etching treatments, was tested. The CNTs were synthesized through a customized thermal CVD chamber enabling rapid (~15 min), cost-effective, and reproducible VA-CNT growth on a whole wafer (2 inch) or on different samples simultaneously. The resulting CNTs were multiwalled carbon nanotubes with diameters ranging between 15 and 25 nm and lengths up to 200 μm, as corroborated by SEM and TEM analyses performed on similar structures synthesized with the same procedure [36,37,38,43]. Various studies, aiming to evaluate the morphology and the degree of alignment of CNT forests through post-processing micrograph analyses or X-ray scattering techniques, have highlighted that these carbon nanostructures are not vertically aligned over their entire length [63,64,65,66]. In particular, a non-aligned crust layer, composed of randomly oriented carbon nanotubes, was observed over the top surface of the VA-CNT forests. The ability of VA-CNTs to modulate bacterial survival is strongly related to the micro- and nano-arrangements of the tubes along their whole length, from the base to the topside. In this direction, we designed experiments related to the modulation of CNTs’ nanomorphology, with the final aim of exposing the sharp tips of the CNTs, considered the nano-features mostly responsible for bacterial wall damage via punctuations [16]. Therefore, since one of the most efficient strategies to tune CNTs’ morphology, without altering their verticality, has turned out to be plasma etching, we investigated the impact of different plasma etching conditions firstly on the tube morphology and topography, via electron and atomic force microscopies, and then on the CNTs’ surface composition by means of XPS characterization. Both the SEM and AFM analyses revealed the formation of spiky CNT bundles after O_2_ plasma etching with an enhanced top-surface roughness when compared to that of the as-grown counterpart. However, the combination of Ar and O_2_ as an etching gas for the treatment gave rise to a less rough VA-CNT surface, thus suggesting the removal of the crust layer as proposed in a previous work [40]. At the same time, the surface composition characterized via the XPS technique revealed a dominant component related to sp^2^-hybridized carbon for the three varieties of the CNT samples. Additionally, a small amount of sp^3^-hybridized carbon was always present, and, in the case of the O_2_-plasma-etched CNTs, it was slightly greater, probably due to the formation of spiky CNT bundles [40,60,61].

Carbon-based nanomaterials have unequivocally demonstrated their ability to kill bacteria [67,68]. Among them, CNTs, especially when dispersed in liquid media, have attracted increased attention as antimicrobial nanotools whose efficiency is highly affected by their purity, size distribution, dispersion state, and media [69]. VA-CNTs are, here, proposed as alternative antibacterial platforms by which it is possible to overcome the main issues related to CNT suspensions and, simultaneously, modulate the survival rate of microorganisms with improved effectiveness and a reduced interaction time. Indeed, different from what was observed previously [16] in the case of CNT suspensions, where antimicrobial effects were visible only after 24 h of treatment, here, a higher bactericidal action of the as-grown or O_2_-etched VA-CNTs was measured after 4 h of treatment and after 30 min in the case of the Ar/O_2_ CNT treatment. An FE-SEM analysis demonstrated that the CNT network observed around the bacterium body caused mechanical injuries to cell walls. This network became denser in the case of the Ar/O_2_-modified CNTs, suggesting that the higher mortality observed in the cell viability tests could be due to the strong interactions with these nanostructures. These observations were most prominent with the Gram-negative *P. aeruginosa*, which possesses thin peptidoglycan layers of about five nm [70]. Since the mechanical damage could be due to interactions with external bacterial structures, the reduced susceptibility of *S. aureus* to VA-CNT deformation might partially be clarified by the increased peptidoglycan thickness, providing an increased rigidity and a higher turgor pressure. Overall, our observations highlight that Gram-negative *P. aeruginosa* was more sensitive than the Gram-positive *S. aureus* strains against the VA-CNTs tested. This behavior has also been observed in antimicrobial tests with nanopillar nanomaterials [70] and, in particular, with CVD-grown VA-CNTs modified or not with plasma etching treatments [71]. Nevertheless, they modified the thickness of CNT forests during growth and their surface chemistry with post-growth treatments by using O_2_ or CF_4_ as an etching gas in order to control the elastic energy stored in the nanotubes and their interaction with bacteria. Additionally, they achieved reduced bactericidal rates (99.3% for *P. aeruginosa* and 84.9% for *S. aureus*) compared with those reached in the present work (100% and 97% for *P. aeruginosa* and *S. aureus*, respectively), where O_2_ or a combination of Ar and O_2_ was exploited to modulate VA-CNTs’ nanomorphology and, consequently, their antimicrobial power without altering their surface compositions. The mechanism of CNTs’ toxicity is highly influenced by several factors, such as their diameter, length, electronic structure, and surface functional group [72]. In general, it has been reported that the antimicrobial activity could be affected by different mechanisms. The first phase includes physical contact between the CNTs and bacteria, following the bacterial membrane perturbation caused by this interaction (phase 2) [73]. The third phase is characterized by electronic structure-dependent bacterial oxidation and subsequent cell membrane damage and death [70]. The first parameter involved is the length of the nanotubes, which influences the interactions with the cell membrane of bacteria. Indeed, shorter tubes seem to exert superior bactericidal performance, increasing the interaction between the open ends of nanotubes and a microorganism [74]. Moreover, smaller diameters can also damage cell membranes, penetrating the interior of the cell and preventing cell proliferation. This induces cell lysis and kills the bacteria. The attachment of CNTs alters the structure, permeability, and proton motive force of the cell membrane. Several studies have shown that the contact of bacteria with CNTs causes cell morphology distortion, damage to cell membrane integrity, and the release of intracellular material [75,76,77]. Although these studies reported that the cytotoxicity of CNTs against bacteria is due to cell membrane damage caused by direct contact, the antibacterial effect could be further related to the production of ROS in CNT-treated cells [78]. In general, ROS, including the superoxide anion (O^2•−^), hydrogen peroxide (H_2_O_2_), hydroxyl radical (^•^OH), and singlet oxygen (^1^O_2_), can be produced within the cell as an unavoidable consequence of bacterial metabolism or derived from the environment [79]. Highly active ROS, especially ^1^O_2_, can directly cause oxidative damage to nucleic acids, proteins, lipids, and other external structures of bacteria, resulting in genotoxicity, protein dysfunction and membrane disruption, and finally bacteria death [80]. On these bases, the high antimicrobial power of the Ar/O_2_-etched VA-CNTs seemed to be due to the synergistic effects of both mechanical injuries and high levels of ROS production. As reported in [73], these metallic nanotubes could induce oxidative stress by acting as a conductive bridge with the bacterial membrane. It has been reported that VA-CNTs demonstrate better field emission than randomly oriented CNTs [81]. Indeed, in the Ar/O_2_-etched VA-CNTs, the crust on top of the forest was removed, allowing them to “short-circuit” the bacteria.

Thanks to their physical properties and morphologies, VA-CNTs are becoming the most attractive nanomaterials for biologically active surfaces. Furthermore, their surface characteristics can be altered through various post-treatment techniques, such as chemical modification and plasma processing. Compared to non-aligned structures, VA-CNT arrays are of specific interest due to their larger available surface area, higher packing density, and controllable microstructure. Finally, controlling the interactions between the surface of nanostructures and cellular organisms can have vital implications for a host of fields, such as in medicine, agrochemical industries, and pharmaceutics. These examples show the usefulness of nanostructures in bio-related applications.

## 5. Conclusions

This study proposes and demonstrates the possibility of designing highly antibacterial coatings with vertically aligned carbon nanotubes, produced by means of an easy, low-cost, and fast (~15 min) chemical vapor deposition process. Different plasma etching treatments were exploited to modulate, up to the sub-micrometric domain, the surface topography of VA-CNTs. Among the different types of developed VA-CNT-based substrates, we found that the combination of Ar and O_2_ as an etching gas gave rise to the most efficient antibacterial power, with an almost complete reduction in bacterial viability after only 30 min of treatment. Additionally, this work sheds light on the mechanism responsible for this exceptional antimicrobial activity of CNT forests, pointing out that it occurs due to the mechanical interaction between the bacterial cell walls and the nanotube structures, which are capable of “skewering” or “smothering” the bacteria and, simultaneously, producing high levels of ROS, thus preventing the formation of microbial colonies. The synthesis process proposed here, involving the CVD growth of CNT forests followed by the subsequent tuning of their nanomorphology through plasma etching, is an effective, rapid (CVD synthesis and plasma etching time is around 20 min overall), and low-cost route for engineering antibacterial films for a wide range of applications, such as biomedical devices, filtering systems for hospitals, and solid–air/liquid interfaces in healthcare units where biofilms usually appear.

## Figures and Tables

**Figure 1 nanomaterials-13-01081-f001:**
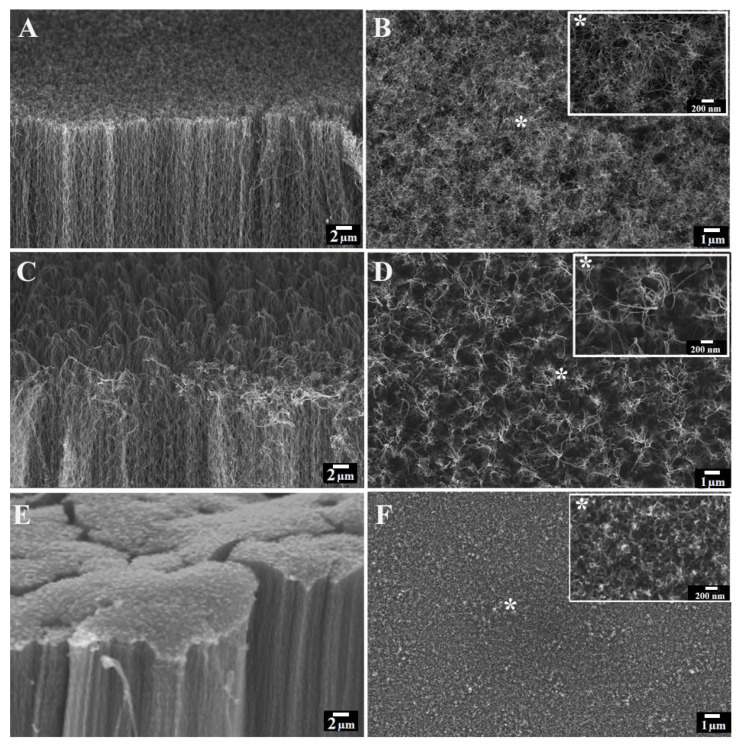
(**A**,**C**,**E**) Side- and (**B**,**D**,**F**) top-view SEM images of as-grown (**top**), O_2_- (**middle**), and Ar/O_2_-plasma-etched (**bottom**) VA-CNTs. Scale bar of star-signed inserts, showing top-view SEM images acquired at increased magnification of as-grown (**B**), O_2_- (**D**), and Ar/O_2_-(**F**) plasma-etched VA-CNTs: 200 nm.

**Figure 2 nanomaterials-13-01081-f002:**
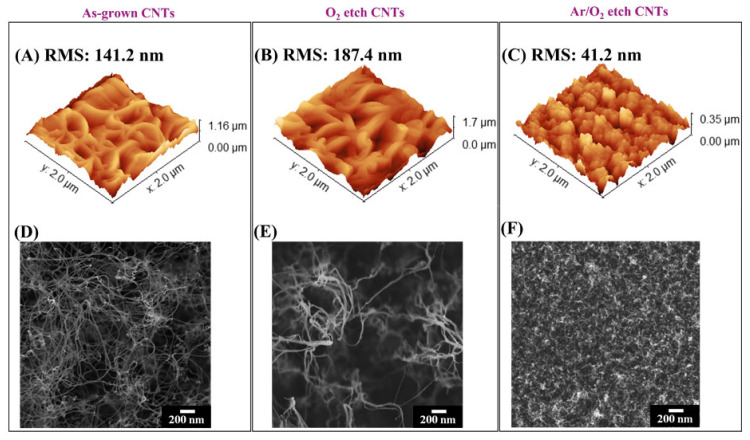
(**A**–**C**) AFM and (**D**–**F**) SEM images of VA-CNTs top surface for (**A**,**D**) as-grown, (**B**,**E**) O_2_-, and (**C**,**F**) Ar/O_2_-plasma-etched. Scale bar: 200 nm. In the top panels, the RMS of the top surface profile is reported.

**Figure 3 nanomaterials-13-01081-f003:**
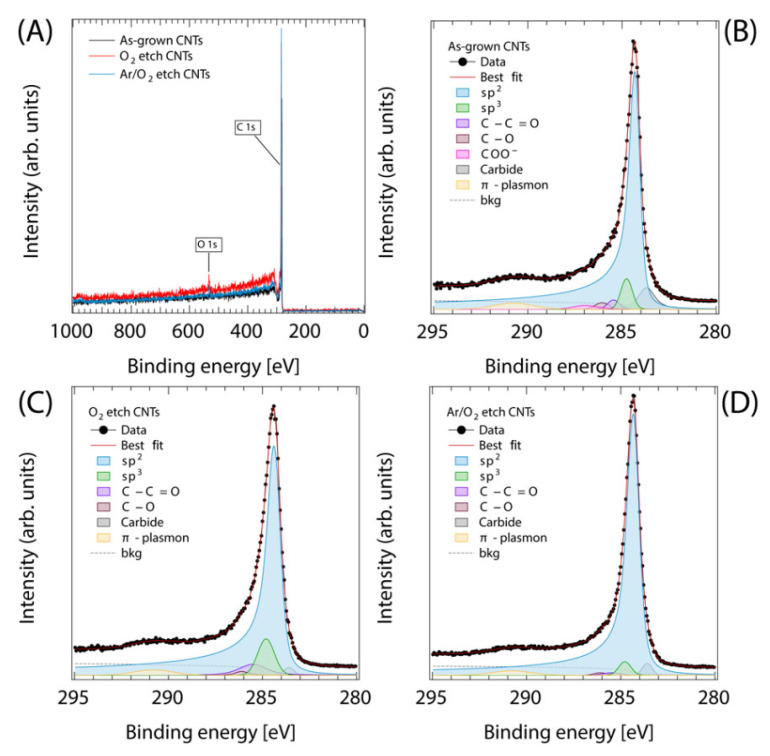
XPS survey (**A**) and C 1s of as-grown (**B**), O_2_- (**C**), and Ar/O_2_-plasma-etched (**D**) VA-CNTs.

**Figure 4 nanomaterials-13-01081-f004:**
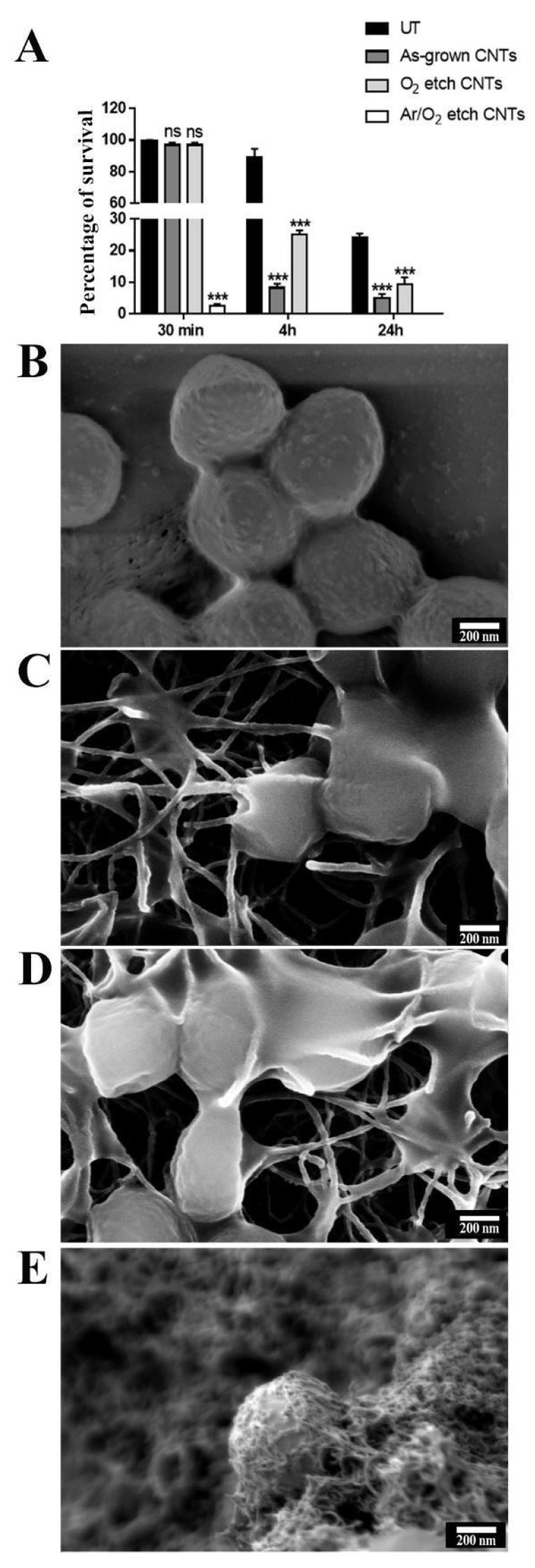
Antibacterial activity of VA-CNTs against *S. aureus*. (**A**) CFU percentage of *S. aureus* after 30 min, 4 h, and 24 h of incubation on Si substrates modified with CNTs. Bars represent the mean of three independent experiments. To assess statistical analysis, a one-way ANOVA analysis with the Bonferroni post-test was used; *** *p* < 0.001 with respect to untreated controls (UT); ns: not significant. The experiment was performed in triplicate for each treatment. SEM micrograph of (**B**) *S. aureus* bacteria incubated on Si substrate without carbon nanostructures in comparison with (**C**) as-grown, (**D**) O_2_, and (**E**) Ar/O_2_. Scale bar: 200 nm.

**Figure 5 nanomaterials-13-01081-f005:**
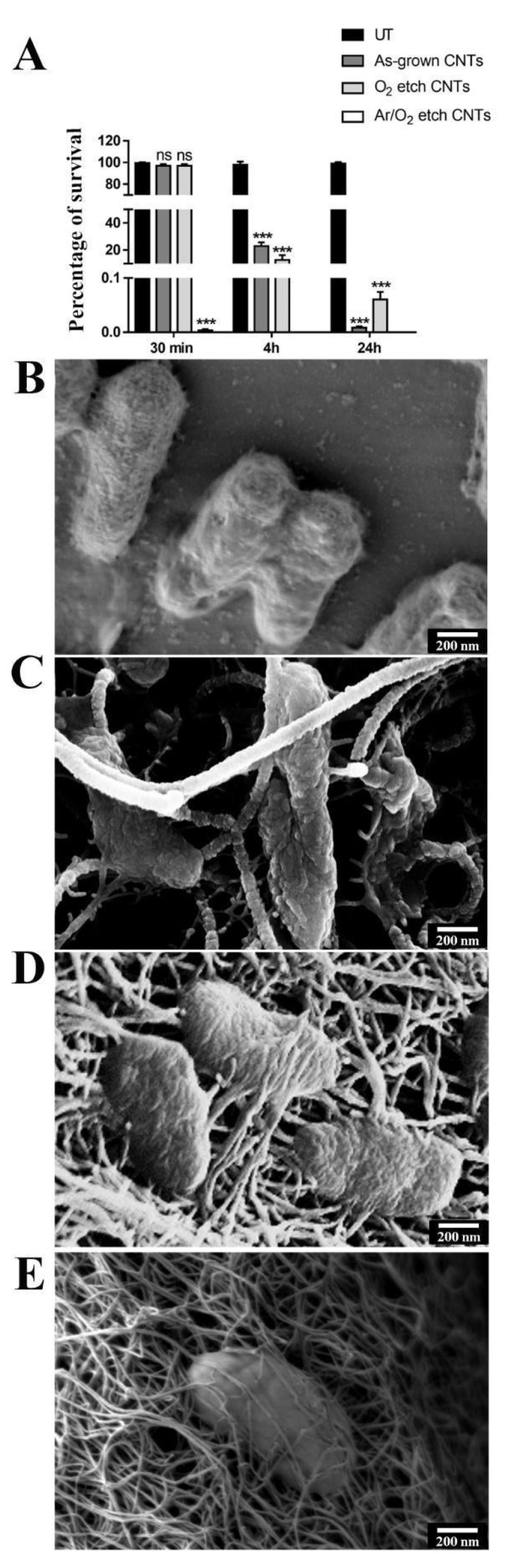
Antibacterial activity of VA-CNTs against *P. aeruginosa.* (**A**) Cell recovery by CFU counting of *P. aeruginosa* cells after 30 min, 4 h, and 24 h of incubation on Si substrates modified or not with different CNTs. Bars represent the mean of three independent experiments. To assess statistical analysis, a one-way ANOVA analysis with the Bonferroni post-test was used; *** *p* < 0.001 with respect to untreated controls (UT); ns: not significant. The experiment was performed in triplicate for each treatment. SEM micrograph of (**B**) *P. aeruginosa* bacteria incubated on Si substrates without carbon nanostructures in comparison with cells treated with (**C**) as-grown, (**D**) O_2_-, and (**E**) Ar/O_2_-plasma-etched VA-CNTs. Scale bar: 200 nm.

**Figure 6 nanomaterials-13-01081-f006:**
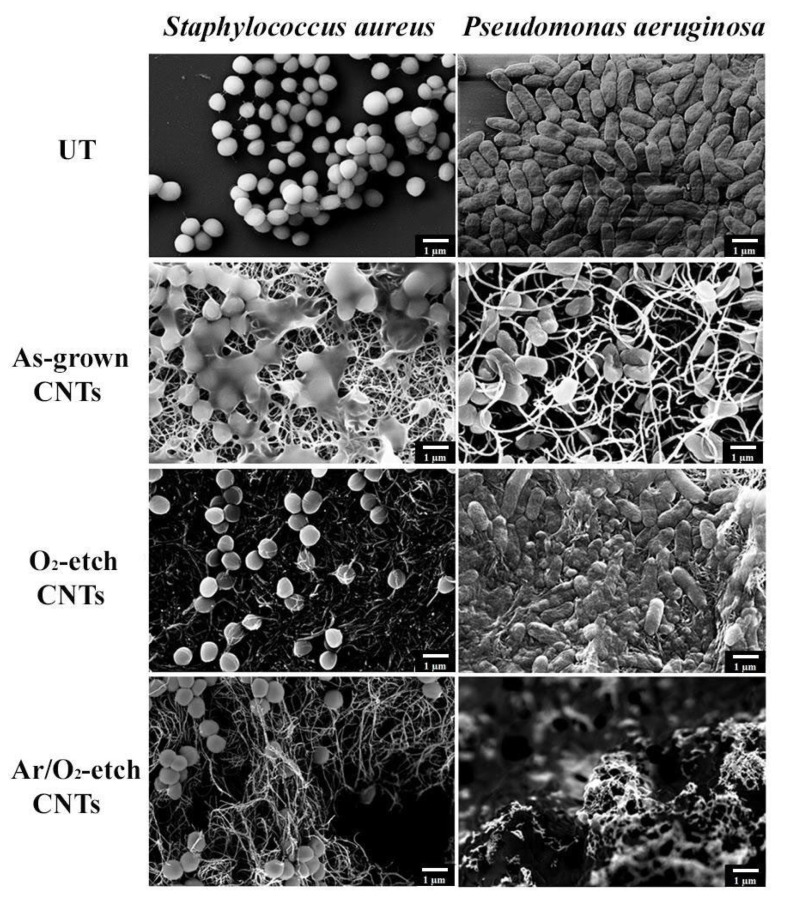
Antibiofilm activity of VA-CNTs against *S. aureus* and *P. aeruginosa*. FE-SEM micrographs of biofilm produced by Gram-positive or Gram-negative bacteria after 24 h incubation on Si substrates modified or not with different CNTs. Untreated (UT) samples were used as control. Bar, 1 µm.

**Figure 7 nanomaterials-13-01081-f007:**
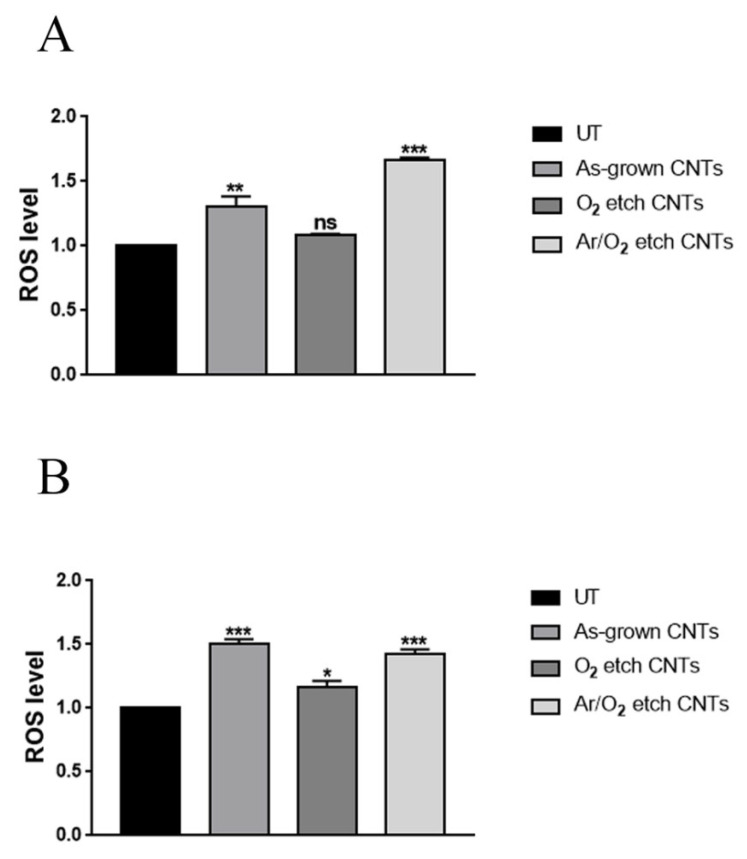
ROS production of (**A**) *S. aureus* and (**B**) *P. aeruginosa* after 2 h incubation on Si substrates modified or not with different CNTs. Untreated (UT) samples were used as control. Bars represent the mean of three independent experiments (* *p* < 0.05; ** *p* < 0.01; *** *p* < 0.001; ns: not significant).

## Data Availability

The data presented in this study are available on request from the corresponding author.

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
