# Peer review of "Plasma-Etched Vertically Aligned CNTs with Enhanced Antibacterial Power"

_nanomaterials, 2023, doi:10.3390/nano13061081_

Round 1
Reviewer 1 Report (Previous Reviewer 4)
The manuscript studied the activity of Staphylococcus aureus and Pseudomonas aeruginosa bacteria placed on the surfaces of three varieties of VA-CNTs: as-grown, and after two different post-growth etching treatments. The effect of Oâ‚‚ or a combination of Ar and Oâ‚‚ with different parameter on topography, antibacterial and antibiofilm activity should be studied deeply to obtain the optimal condition of plasma etching processes, which can be used to modulate VA-CNTs nano-morphology and their antimicrobial power. In order to further investigate the surface chemical reaction occurring between CNT forests and Oâ‚‚ or Ar/Oâ‚‚ etching gas, more characterizations besides XPS characterization should be performed and more convincing analysis and discussion should be presented. In addition, the reason for inducing variations of ROS levels production should be investigated.
Author Response
The authors thank the reviewer for the feedback and the valuable comments.
Please see the attachment for the point-by-point reply.

Reviewer 2 Report (New Reviewer)
This study proposes and demonstrates the possibility of designing highly antibacterial coatings with vertically aligned carbon nanotubes, produced by means of an easy, lowcost and fast (~15 min) chemical vapor deposition process. One of the novelties of the work is what it sheds light on the mechanism responsible of this exceptional antimicrobial activity of CNT forests, pointing out that it occurs due to the mechanical interaction between the bacterial cell walls and the nanotube structures, which are capable of 'skewering' or 'smothering' the bacteria and, simultaneously, producing high levels of ROS, thus preventing the formation of microbial colonies.
Three different varieties of VA-CNTs were investigated, in terms of antibacterial and antibiofilm activity, against Pseudomonas aeruginosa and Staphylococcus aureus: as-grown, and after two different etching treatments.
S. aureus. and P. aeruginosa are one of highly virulent bacterial pathogens. It is part of ESKAPE pathogens. ESKAPE is an acronym for six highly virulent bacterial pathogens including: Enterococcus faecium, Staphylococcus aureus, Klebsiella pneumoniae, Acinetobacter baumannii, Pseudomonas aeruginosa, and Enterobacter cloacea. These pathogens are multidrug resistant and are the main causative agents of infections, including nosocomial infections.
The manuscript is written clearly and understandably without frills. All conclusions supported by the results. The paper can be published in Nanomaterials after minor revision.
Why the authors limited their research to S. aureus. and P. aeruginosa?
Author Response
The authors thank the reviewer for the feedback and the valuable comments.
Please see the attachment with the point-by-point reply.

Reviewer 3 Report (New Reviewer)
This research reports on the preparation of vertically-aligned CNTs by catalytic chemical vapor deposition, optionally followed by plasma etching process. The performance of CNTs as antibacterial agent against Pseudomonas aeruginosa and Staphylococcus aureus are investigated. The results obtained can add to the body of the literature. However, the following issue should be addressed before publication:
1. Introduction: “This results in the emergence of multidrug-resistant (MDR) bacteria [2], an increasing trend due to the amplified use of antibiotics, especially during the coronavirus pandemic [3].”
This is statement is not accurate since antibiotics are not used against virus species. Also, Reference [3] “Synthesis of ZnO/Au Nanocomposite for Antibacterial Applications”, which is used to support the statement does not provide any statistics to prove the claim.
2. The manuscript can be enriched by recent articles published on the antibacterial performances of carbon nanostructures:
- 3D-graphene nanosheets as efficient antibacterial agent, Materials Letters 321 (2022) 132406.
- The Antibacterial Properties of Nanocomposites Based on Carbon Nanotubes and Metal Oxides Functionalized with Azithromycin and Ciprofloxacin, Nanomaterials 2022, 12(23), 4115
- Dispersant Effects on Single-Walled Carbon Nanotube Antibacterial Activity, Molecules 2022, 27(5), 1606
3. Experimental Section: “… for the deposition of a thin (3 nm in thickness) catalyst layer of iron over the Si-based substrates”. How did author measure the thickness? Please add reference if it has been discussed in previous publications.
4. Experimental Section: “…and annealed at 720 °C in Hâ‚‚ atmosphere” Did the author used pure H2 atmosphere? If yes, the safety issues regarding the use of such explosive gas must be mentioned.
5. Experimental section: “The growth time, intended as the time of interaction between iron nanoparticles” This statement applies that iron deposited in the substrate is the form of nanoparticles. Is his correct? If yes, what are the size and morphology of such nanoparticles? Are there any microscopy evidence for this?
6. Sections 2.1, 2.2 and 2.3 discuss the experimental procedures employed for the production of characterization of CNTs. These sections, which are reported, based on pervious publications are in great details. However, section 2.4 that deals with the bacteria growth conditions and cell viability test is very short. This section must be completed with more details to ensure the repeatability of the test.
7. Are CNTs single-walled or multi-walled?
8. Page 8, the discussion made on XPS results must be supported by the literature.
Author Response
The authors thank the reviewer for the feedback and the valuable comments.
Please see the attachment with the point-by-point reply.

Round 2
Reviewer 1 Report (Previous Reviewer 4)
“The effect of Oâ‚‚ or a combination of Ar and Oâ‚‚ with different parameter on topography, antibacterial and antibiofilm activity should be studied deeply to obtain the optimal condition of plasma etching processes, which can be used to modulate VA-CNTs nano-morphology and their antimicrobial power”.
The comment is not addressed. The manuscript mainly studied the activity of Staphylococcus aureus and Pseudomonas aeruginosa bacteria placed on the surfaces of three varieties of VA-CNTs: as-grown, and after two different post-growth etching treatments. In the authour’s previous works, they have characterized the VA-CNT-based substrates via a combination of microscopic (SEM, TEM, AFM) and spectroscopic (XPS and Raman) techniques [36-38,43]. HOWEVER, The effect different parameter on antibacterial and antibiofilm activity was not be studied deeply to obtain the optimal condition of plasma etching processes, which can be used to modulate VA-CNTs antimicrobial power. After the problems which are potentially of interest to the readership are addressed, the manuscript can meet the required quality standards to be considered for publication. Accordingly, the reviewer does not recommend publication of the present version of the paper in the Nanomaterials.
Author Response
Please see the attachment

This manuscript is a resubmission of an earlier submission. The following is a list of the peer review reports and author responses from that submission.
Round 1
Reviewer 1 Report
Schifano et al reported the plasma etched vertically aligned CNTs for antibacterial purpose. The design of this project is very interesting, and very appealing for the people who have been suffered for bacterial issues. It might be inspiring for people and country which were searching such kind of techs for prohibiting bacterial. So, it will be interesting for readers from Nanomaterials. The paper was organized well, and all results have been properly characterized. It could be accepted at current status.
Reviewer 2 Report
Dear Authors, I really like your manuscript and recomend to publish it.
Reviewer 3 Report
In this study, the plasma etched vertically aligned CNTs with enhanced antibacterial power were investigated.
However, the main problem of this study is the lack of discussion and novelty.
The proposed method is different from the conventional CNT, indicating the dispersion is not important. Thus, the authors should compare the proposed CNT with conventional CNT and emphasize the advantages of using the proposed CNT regarding the dispersion problem.
In addition, more experiments and analysis are required to observe the 'antibacterial power' as mentioned in the title.
For these reasons, the reviewer suggests reject.
Reviewer 4 Report
In the work, the antimicrobial properties of VA-CNT forests was investigated. There are some problems as follows.
(1) The novelty of this manuscript is not evident. The method for preparing Vertically-aligned carbon nanotubes is a mature technology. And the post treatment is an accessible process. The manuscript mainly studied the activity of Staphylococcus aureus and Pseudomonas aeruginosa bacteria placed on the surfaces of three varieties of VA-CNTs: as-grown, and after two different post-growth etching treatments. However, the VA-CNTs obtained by the post-growth etching treatments were not characterized and analyzed in-depth. The activity of Staphylococcus aureus and Pseudomonas aeruginosa bacteria is simply evaluated by CFU counting analysis and morphology.
(2) Abstract
The structure of the abstract should be rearranged to demonstrate the nature, importance, and contributions of the paper. It should include more important finding and results.
(3) Conclusion
The conclusion should summarizes some of the paper's most important ideas without long descriptions of backgroud, method, etc.
(4) The font size in the figures should be same. Some text in the figures is too small to be seen. The scale ruler in inset in Figure 1, Figures 4, 5 and 6 should be clearly shown. CFU percentage of S. aureus in Figures 4, 5 should be redrawn and clearly demonstrated.
Above all, I think the manuscript in its current form is not suitable for publishing in Nanomaterials. Authors should conduct relative characterizations and do further analysis to improve the manuscript.